# Altered kinetics of circulating progenitor cells in cardiopulmonary bypass (CPB) associated vasoplegic patients: A pilot study

Sanhita Nandi[1☯], Uma Rani Potunuru[2☯], Chandrani Kumari[3], Abel Arul Nathan[1], Jayashree Gopal[4]*, Gautam I. Menon[3,5], Rahul Siddharthan[3], Madhulika Dixit[1]*, Paul Ramesh Thangaraj[6,7]*

1 Laboratory of Vascular Biology, Department of Biotechnology, Bhupat and Jyoti Mehta School of Biosciences, Indian Institute of Technology Madras, Chennai, India, 2 Apollo Hospitals Educational and Research Foundation, Chennai, India, 3 The Institute of Mathematical Sciences (HBNI), Chennai, India, 4 Department of Endocrinology and Diabetology, Apollo Hospitals, Chennai, India, 5 Departments of Physics and Biology, Ashoka University, Sonepat, India, 6 Department of Cardiothoracic Surgery, Apollo Hospitals, Chennai, India, 7 Department of Mechanical Engineering, Indian Institute of Technology Madras, Chennai, India

☯ These authors contributed equally to this work.
* jgopal25268@gmail.com (JG); mdixit@iitm.ac.in (MD); paulramesh@gmail.com (PRT)

**Data Availability Statement:** All relevant data are within the manuscript and its Supporting Information files.

## Abstract

Vasoplegia observed post cardiopulmonary bypass (CPB) is associated with substantial morbidity, multiple organ failure and mortality. Circulating counts of hematopoietic stem cells (HSCs) and endothelial progenitor cells (EPC) are potential markers of neo-vascularization and vascular repair. However, the significance of changes in the circulating levels of these progenitors in perioperative CPB, and their association with post-CPB vasoplegia, are currently unexplored. We enumerated HSC and EPC counts, via flow cytometry, at different time-points during CPB in 19 individuals who underwent elective cardiac surgery. These 19 individuals were categorized into two groups based on severity of post-operative vasoplegia, a clinically insignificant vasoplegic Group 1 (G1) and a clinically significant vasoplegic Group 2 (G2). Differential changes in progenitor cell counts during different stages of surgery were compared across these two groups. Machine-learning classifiers (logistic regression and gradient boosting) were employed to determine if differential changes in progenitor counts could aid the classification of individuals into these groups. Enumerating progenitor cells revealed an early and significant increase in the circulating counts of CD34$^+$ and CD34$^+$CD133$^+$ hematopoietic stem cells (HSC) in G1 individuals, while these counts were attenuated in G2 individuals. Additionally, EPCs (CD34$^+$VEGFR2$^+$) were lower in G2 individuals compared to G1. Gradient boosting outperformed logistic regression in assessing the vasoplegia grouping based on the fold change in circulating CD 34$^+$ levels. Our findings indicate that a lack of early response of CD34$^+$ cells and CD34$^+$CD133$^+$ HSCs might serve as an early marker for development of clinically significant vasoplegia after CPB.

**Funding:** 1)URP received National Post-Doctoral fellowship File No. PDF/2017/002087, SERB-DST, Government of India (https://serbonline.in/SERB/HomePage). 2) MD received (a) Institute Research & Development Award (IRDA): (IRA/14-15/003/RESF/MADH); IIT Madras; (https://www.cse.iitm.ac.in/awards_details.php?arg=NDQ=) (b)Impacting Research Innovation & Technology (IMPRINT): (35-1816/2016-T.S.-1); Ministry of Human Resource Development, Govt. of India (https://www.imprint-india.org/) (c) Exploratory Research Project (ERP): (BIO/18-19/873/RFER/MADH) (https://mhrd.gov.in/). 3) CK, GIM and RS acknowledge funding from the Computational Biology project at their institute funded by the Department of Atomic Energy, Government of India.(http://dae.gov.in/) NO- The funders had no role in study design, data collection and analysis, decision to publish, or preparation of the manuscript.

**Competing interests:** Authors have declared that no competing interests exist.

**Abbreviations:** CPB, Cardiopulmonary bypass; HSCs, Hematopoietic stem cells; EPC, Endothelial progenitor cells; NE, Norepinephrine; FITC, Fluorescein isothiocyanate; APC, Allophycocyanin; ICU, Intensive care unit; MAP, Mean Arterial Pressure; SBP, Systolic blood pressure; DBP, Diastolic blood pressure; CART, Classification And Regression Tree.

# 1. Introduction

Vasoplegia is a well recognized post-surgical complication of cardiopulmonary bypass (CPB) [1,2]. Endothelial activation and dysfunction are considered to be pivotal in the pathophysiology of vasoplegia. Kortekaas *et al* reported that pre-existing endothelial cell activation, determined by increased levels of von Willebrand factor (vWF) and soluble P-selectin at baseline, predisposes patients to vasoplegia [3]. Brettner et al have suggested that deviations in endothelial injury markers in a CPB group, as compared to an off-pump surgery group, identifies a correlation between ischemia/reperfusion and the extent of endothelial activation [4]. Circulating endothelial cells have also been reported to be elevated at baseline in cardiac surgery patients compared to healthy individuals, with their numbers further increasing post CPB [5]. Additionally, impairment in endothelial barrier function is observed during the course of cardiac surgery and post-surgery, when assessed through the prism of the angiopoetin-Tie2 system [6–8].

Though multiple studies suggest that a compromised endothelium along with continued vascular damage during CPB contributes to vasoplegic syndrome, the status of vascular repair and vascular progenitors in these settings is ill-characterized. Circulating haematopoietic stem cells (HSCs: CD34$^+$CD133$^+$) and endothelial progenitor cells (EPCs: CD34$^+$VEGFR2$^+$) are known to home in for repair, to the site of damaged endothelium and the vessel wall [9]. Reduced numbers and an altered function of EPCs are associated with ischemic cardiovascular diseases [10,11]. However, it is not clear how their circulating counts vary during the unfolding of the vasoplegic syndrome in CPB.

We sought to determine the changes in the circulating counts of these progenitors and their relationship to the severity of vasoplegia in cardiac patients undergoing CPB. We also trained machine learning methods (logistic regression and gradient boosting) to assess the accuracy of the vasoplegic categorization of patients based on these markers.

# 2. Materials and methods

## 2.1 Materials

Vacutainers for blood collection K2-EDTA (Cat: 367863) and serum separation (Cat: 367812) were purchased from Becton Dickinson Biosciences. Fluorescent antibodies such as anti-human CD133-Phycoerythrin (PE) (Cat: 130-080-801) were purchased from Miltenyi Biotech Anti-human CD34-Fluorescein isothiocyanate (FITC) (Cat: 343604), anti-human VEGFR2-Allophycocyanin (APC) (Cat: 359916) and isotype control antibodies like mouse IgG2a-FITC (Cat: 400207) and mouse IgG1k-PE (Cat: 400113), were purchased from BioLegend. Other dry chemicals were purchased from Sigma-Aldrich.

## 2.2 Surgical procedure

Cardiac surgery was performed according to local standardized protocols. All surgical procedures were performed via a midline sternotomy under normothermic CPB. Blood cardioplegia was used for all the cardiac cases. The cardioplegic solution was a mixture of patient's own blood and crystalloid solution. The crystalloid solution was a mixture of Plasmalyte (500 mL) and St. Thomas' solution (20 mL). The delivery was done by the use of a specialized roller pump (Sarns Medical Systems) at the ratio of 4:1 where 4 parts of blood and 1 part of the mixed cardioplegia solution were used. In lung transplant the lung preservative used was Perfadex solution (Medisan, Uppsala, Sweden) containing low-potassium (K+ 6 mmol/L) and extracellular electrolytes (Na+ 138 mmol/L; Cl- 142 mmol/L; Mg++ 0.8 mmol/L; Dextran 40 g); H2PO4- 0.8 mmol/L; Glucose (0.91 g); Osmolarity (292 mOsm/L).

## 2.3 Study design and sample collection

Following informed consent, subject recruitment and sample collection were done at the Cardiothoracic Surgical unit of Apollo Hospitals, Chennai. The study design was reviewed and approved by the Institute Ethics Committees of Apollo Hospital [AMH-001/01-18] as well as the Indian Institute of Technology Madras [IEC# IEC/2016/02/MDX-2/05]. A total of 19 elective cardiac surgery individuals (8 male and 11 female), age range 40–70 years (adults), and undergoing CPB, were recruited. Written consent as prescribed by the institutional ethics committees was obtained from patients a day prior to their operation. Individuals with emergency trauma surgery or any infectious diseases were excluded.

The blood samples were collected in $K_2$-EDTA vacutainers for subsequent flow cytometry from a central venous catheter placed in the jugular vein when the patient was not on the CPB pump. When the patient was on pump, blood was collected from the inlet of the pump at specified time intervals as described below. The whole blood collected was processed within 2 hours of collection for flow cytometry. The time points for blood collection were: (1) 'Post-Induction' (time after the induction of anesthesia prior to the start of surgery); (2) 'Sternotomy' (beginning of surgery after median sternotomy); (3) '0 hour' (start of the CPB pump); (4) '1 hour' (1 hour on pump); (5) '6 hour'(end of surgery when the patient was shifted to ICU) (6) '24 hour' (time point when the patient is in ICU for recovery). These time points and study design are summarized in Fig 1. To minimize inter patient bias which may contribute to variability, we considered each patient's baseline i.e. 'Post-Induction' as the control for that patient.

## 2.4 Grouping of subjects based on severity of vasoplegia

In the absence of a standard consensus for the definition of clinical vasoplegia [12,13], we choose to categorize our recruited subjects based on the total dose of norepinephrine (NE) infused in these subjects. In most cardiothoracic units, low levels of NE (usually total dose <0.02mg/kg body weight), are given in all patients post-surgery. This dosage is increased only if a patient further develops vasoplegia. We thus reasoned that increased use of pressor beyond 0.02mg/Kg body weight would identify clinical worsening of vasoplegia. Hence, we choose a cutoff dose of total NE above 0.02mg/kg body weight to group the subjects into two groups. Among the recruited 19 subjects, 11 subjects received NE <0.02mg/Kg body weight and were classified as a clinically insignificant vasoplegia group or group 1 (G1). The remaining 8 subjects who received NE >0.02mg/Kg body weight were termed as a clinically significant vasoplegia group or group 2 (G2).

## 2.5 Enumeration of circulating progenitor cells via flow cytometry

For enumeration of circulating vascular progenitors in whole blood, cells were fixed and erythrocytes lysed using fluorescence-activated cell sorter (FACS) lysing solution. Blocking was performed with 5% Fetal Bovine Serum (FBS) and 0.2% Bovine Serum Albumin (BSA) for 30 min followed by staining with CD133-PE, CD34-FITC and VEGFR2-APC antibodies for 40 minutes at 4˚C. Corresponding isotype immunoglobulin G (IgG) 1-PE and IgG2a-FITC and IgG2aκ-APC antibodies were used as controls. A minimum of 500,000 events were acquired and scored as per EUROSTAR guidelines using a FACS Aria flow cytometer (Becton Dickinson) [14]. Data were analyzed using the Flowjo software program (Version 7.6.1, Tree Star Inc.). Following appropriate gating, $CD34^+$, $CD133^+$, $VEGFR2^+$, EPC ($CD34^+VEGFR2^+$), HSC ($CD34^+CD133^+$) and $CD133^+VEGFR2^+$ cells (S1 Fig) were enumerated from lympho-monocyte fraction and are represented as the number of cells per million lympho-monocytic events.

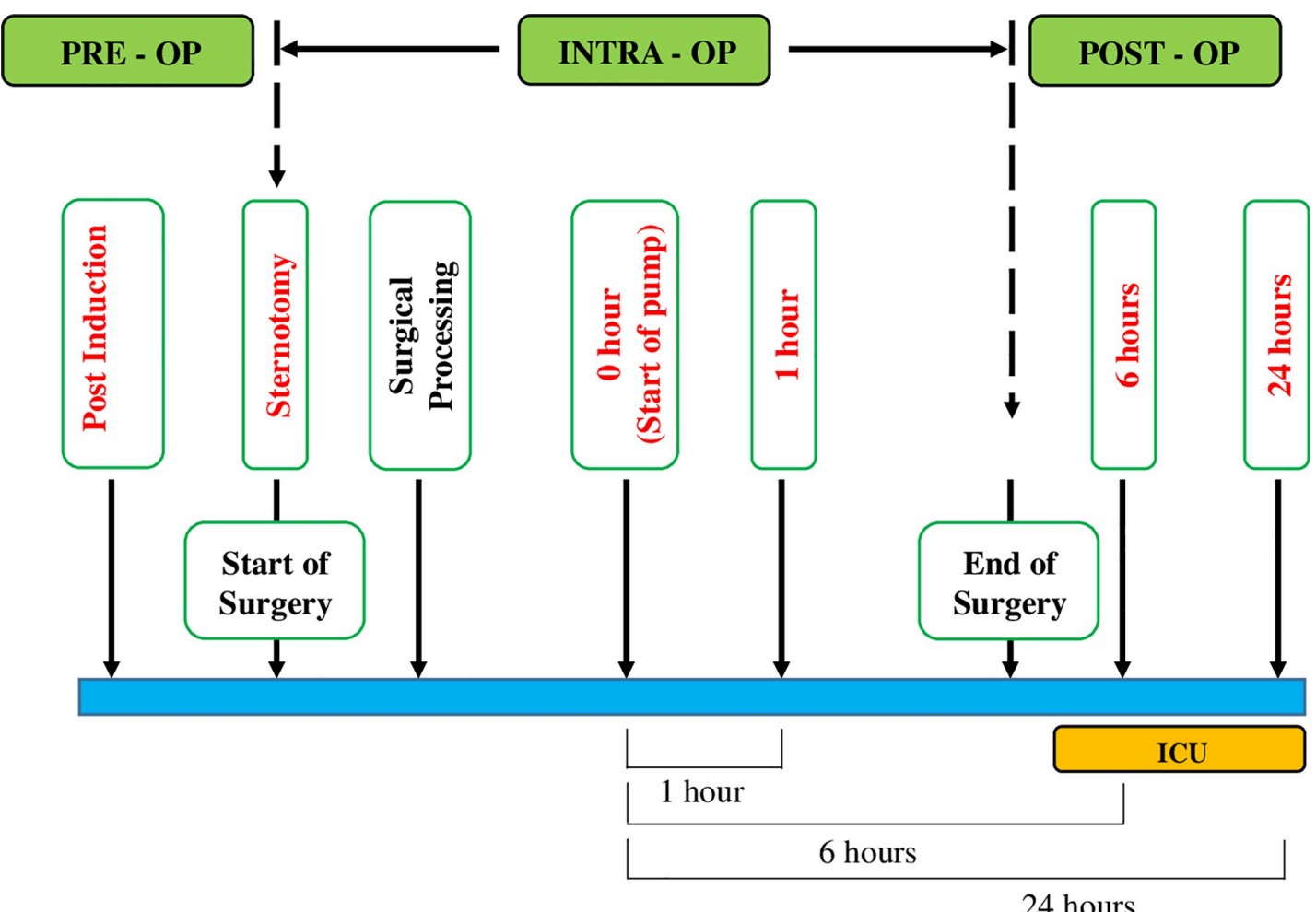

**Fig 1. Study design showing time points of sample collection.**

## 2.6 Statistical analysis of clinical data

Progenitor cell counts are expressed as mean ± SEM of the number of cells counted per 1 million lymphomonocyte events. Clinical data are also expressed either as mean ± SEM or Median. That the data were normally distributed was confirmed through a Kolmogorov-Smirnov test. Comparisons within and across groups were performed using paired and unpaired Student's t-test, respectively. For non-parametric data, the Mann-Whitney test and Wilcoxon matched pair test were employed.

## 2.7 Mathematical model to assess the characterization of study subjects

We used two machine learning techniques, logistic regression and gradient boosting, as prediction tools. Logistic regression is a machine learning tool used for binary classification problems. In linear regression, a linear function of inputs is fitted to match output on training data. The output is a real number.

In logistic regression, the output value is transformed using a cost function known as a logistic or sigmoid function to return a value between zero and one, which can be interpreted

as a probability value. This output can be thresholder to predict a value of 0 or 1. We used the implementation in scikit-learn [15].

Gradient boosting is a decision-tree-based algorithm for classification problems [16]. A decision tree is a tree of possible choices to be made given the input data. In a classification and regression tree (CART), unlike a traditional decision tree, the leaves indicate possible scores for outcomes, rather than binary choices. In gradient boosting, an ensemble of trees is used where each tree is trained to minimize errors of preceding trees. This has become a very popular machine-learning tool for tabular data in recent years, here we use the XGBoost package [17].

Based on input parameters, logistic regression and gradient boosting were used to predict whether a patient belongs to G1 or G2. Predictions were validated using "leave one out"—each patient was left out once, and the machine-learning model was trained on the remaining patients and tested on the left-out patient. The significance of these predictions was assessed using *p*-values, i.e. the probability of a similar or better fit happening by chance. To calculate these, we compared the results of 10,000 randomizations. Random data were generated by shuffling the vasoplegia categorization (that is, redistributing the category labels among the patients, ensuring that there are still the same number of patients as are there in G1 and G2. Here, the *p*-value is the fraction of random data-sets with a similar or lower mean absolute error (MAE; the average of the absolute value of the error over all individuals) to the real data. Similar results (not shown) were obtained with a more aggressive randomization, where for each patient, we retain the vasoplegia categorization but pick a random number between the minimum and maximum of the original graph as the log fold-change value.

## 3. Results

### 3.1 Characteristics of study subjects

A drop in the Mean Arterial Pressure (MAP) from pre-CPB to post-CPB despite administration of inotropes and presence of hyper-lactemia indicates the onset of vasoplegia [1,18]. Hence, we measured intra-arterial systolic blood pressure (SBP), diastolic blood pressure (DBP) and MAP in all 19 study subjects. We observed a significant decrease, not only in the MAP post-CPB at 24 hours but also in DBP at 24 hours compared to the pre-surgery state (Fig 2A). Additionally, levels of lactate in blood were significantly higher after 1 hour on the pump, peaking at 6 hours and remaining significantly high even at 24 hours compared to post-induction (Fig 2B). These parameters were similar for all 19 subjects in our study group, except for the difference in dosage of noradrenaline or norepinephrine (NE) infusion post-CPB.

Table 1 shows the baseline anthropometric and clinical characteristics and intake of medication such as Angiotensin II converting enzymes inhibitors (ACEI) and Angiotensin receptor blockers (ARBs) of the clinically classified groups G1 and G2. As shown in Table 1, there was no significant difference in age, gender ratio, BMI, heart and respiratory rate, DBP, SBP, and MAP as well as in the % intake of ACEIs, ARBs, statins, steroids except for beta-receptor blockers (β- blockers) between the two groups. The intake of β- blockers is almost two fold in G2 subjects. The percentage of existent comorbidities in recruited subjects such as diabetes, hypertension, dyslipidemia and renal failure, along with cardiac morbidities such as heart failure, right ventricular dysfunction (RV dysfunction) and pulmonary artery hypertension (PAH) are shown in Table 1. The percentage of RV dysfunction in G2 subjects is higher compared to G1 subjects. A significant increase in urea for G2 is in concordance with observed renal failure in 25% subjects of this group.

The post-operative clinical outcomes for the two groups are shown in Table 2. The cardiac function measured as left ventricular ejection fraction (LVEF) post CPB, showed normal

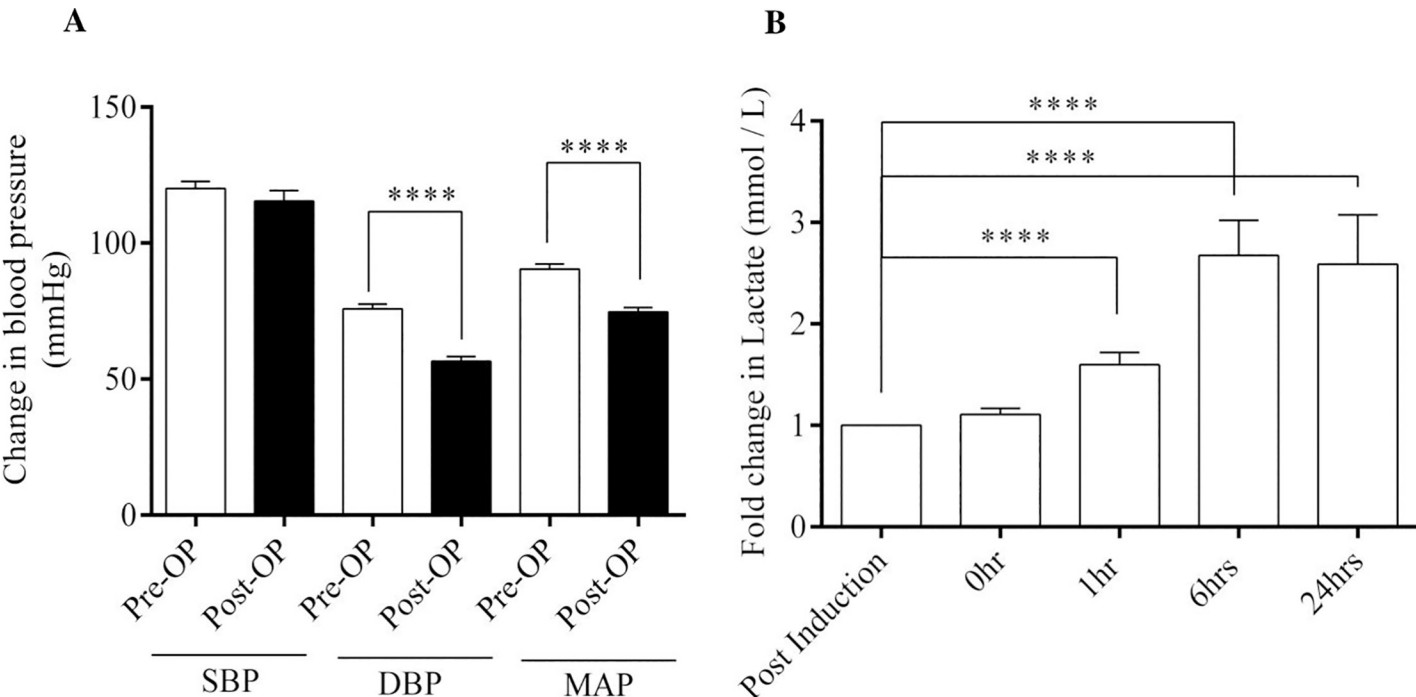

**Fig 2. Characterization of study subjects.** (A) Graph showing blood pressure changes between Pre-Operation and Post-Operation in all the study individuals (n = 19). (B) Lactate concentration from blood at different time points in all the individuals (n = 19) measured in mmol/L. Data shown as Mean ± SEM. ** p ≤ 0.01, ***p ≤ 0.001, **** p ≤ 0.0001 vs respective control. Mann-Whitney's test and Wilcoxon matched pair test were employed.

values of more than 55% for both the groups (Table 2). However, MAP was maintained above 70mmHg for both groups with infusion of vasopressors (Table 2). Intraoperative parameters such as fluid balance have an effect on the post CPB outcomes like vasoplegia, organ dysfunction and mortality [1,19]. Hence, we determined the intraoperative fluid balance change (FBC) during CPB in terms of the difference in volume of fluid intake and output (I & O) for each subject [20] (calculated from the perfusionist chart). The fluid balance change is expressed as positive, negative or zero fluid balance and the percentage of subjects under each fluid balance state in a group is shown in Table 2. The percentage of positive fluid balance subjects is higher in group 2 (75%) compared to group 1 (36.4%). (A positive fluid balance, also known as fluid overload, is correlated to higher mortality and organ dysfunction [21,22]). Moreover, neither group had a trigger Hematocrit (Hct) for transfusion, (Median %Hct Group 1 = 34; Group 2 = 28). The higher incidence of positive fluid balance in G2 reflects the higher incidence of vasoplegia in that group, although it is difficult to establish causality.

There was a significant difference in post CPB characteristics such as the duration of ventilation (p<0.01) and the duration of ICU (p = 0.056) and hospital stay (p<0.05) which were all higher in case of G2 subjects, confirming the severity of clinical vasoplegia in G2 compared to G1 subjects. Among the measured biochemical parameters there was a significant increase in WBC count and a significant decrease in lymphocyte count for G2 compared to G1 subjects (Table 2).

The predictive risk assessment score (EuroSCORE II, SOFA II score and APACHE II) is provided in Tables 1 and 2. The EuroSCORE II (European System for Cardiac Operative Risk Evaluation II) score, used as a pre-operative index of patients' risk profile [23,24] in terms of estimated percentage mortality (% mortality), is shown in Table 1. This score takes into account patient related factors (anthropometric and comorbidities), cardiac related factors

**Table 1. Anthropometric and clinical characteristics of CPB individuals at baseline.**

| Parameters | Group 1 (n = 11) | Group 2 (n = 8) |
|---|---|---|
| Age (yrs) | 57 ± 2 | 54 ± 2 |
| Sex (M/F) | 5/6 | 3/5 |
| BMI | 25.51± 1.02 | 26.46 ± 2.38 |
| Heart Rate (per min) | 83.45 ± 3.32 | 89 ± 6.27 |
| Systolic BP (mmHg) | 121.8 ± 2.26 | 115.0 ± 4.62 |
| Diastolic BP (mmHg) | 78.18 ± 2.26 | 72.5 ± 2.50 |
| Mean Arterial Pressure (mmHg) | 92.59 ± 2.53 | 86.67 ± 2.88 |
| Respiratory Rate (No. of breaths/min) | 20.18 ± 0.68 | 22.13 ± 1.23 |
| LVEF | 61.3±2.9 | 61.7±1.9 |
| Urea (mg/dl) | 24.80 ± 2.19 | 45.29 ± 6.7[**] |
| Creatinine (mg/dl) | 0.74 ± 0.05 | 0.94 ± 0.12 |
| **Comorbidities % prevalence** | | |
| Diabetes (%) | 45.0 | 37.0 |
| Hypertension (%) | 45.0 | 25.0 |
| Dyslipidemia (%) | 9.0 | 12.0 |
| Renal failure (%) | 0 | 25.0 |
| Heart failure (%) | 27.27 | 37.5 |
| Right Ventricular dysfunction (%) | 9.09 | 37.5 |
| PAH (%) | 36.36 | 50.0 |
| **Medication intake (% per group)** | | |
| ACEI (%) | 27.27 | 35.1 |
| ARB (%) | 27.27 | 12.5 |
| Statins(%) | 27.3 | 25 |
| Steroids (%) | 18.2 | 37.5 |
| Beta blockers (%) | 36.4 | 75 |
| Types of surgery | No. of patients | |
| Mitral valve replacement/Repair | 2 | 5 |
| Aortic valve replacement/Repair | 2 | 1 |
| Double valve repair | 1 | 1 |
| Two procedure (CABG+ Valve replacement) | 3 | 0 |
| Double lung transplant | 3 | 0 |
| Double lung transplant+CABG | 0 | 1 |
| Pre operative risk score | | |
| Euro SCORE II (Median % estimated mortality) | 1.25 | 3.05 |

LVEF—Left ventricular ejection fraction, Intake of Angiotensin II converting enzymes inhibitor (ACEI) and Angiotensin receptor blocker (ARB) are expressed in percentage, CABG- Coronary artery bypass graft; Euro SCORE II- European System for Cardiac Operative Risk Evaluation II; Data expressed in Mean± S.E.M unless otherwise specified [**]$p < 0.0$.

(pre-operative LV function, recent myocardial infarction and pulmonary hypertension) and operation related factors (urgency and weight of intervention). The SOFA II (Sequential organ Failure Assessment II) score is used to determine level of organ dysfunction and mortality risk in ICU patients (Table 2). APACHE II (Acute Physiology and Chronic Health Evaluation II) score estimates ICU mortality (Table 2) [25–27]. The categories of post-operative ICU parameters taken into account during calculation of SOFA II and APACHE II are vitals (temperature, mean arterial pressure, heart rate, respiratory rate) and blood chemistry (urea, creatinine, bilirubin and

**Table 2. Clinical and Biochemical characteristics of individuals at intra and post-CPB.**

| Post-CPB (at ICU) measurements | | | |
|---|---|---|---|
| **Clinical parameters** | | **Group 1 (n = 11)** | **Group 2 (n = 08)** |
| CPB duration (min) (Median) | | 225.2 ± 41.04 | 206.1 ± 45.09 |
| Cross Clamping time (min) (Median) | | 198.8 ± 45.64 | 161 ± 45.73 |
| Complete flow rate (lit/min) (Median) | | 3.90 ± 0.16 | 3.97 ± 0.17 |
| Average flow rate(lit/min) (Median) | | 2.89 ± 0.12 | 2.77 ± 0.14 |
| Heart Rate (per min) at 24 hrs | | 77.82 ± 4.14 | 95.00 ± 5.57 * |
| Systolic BP (mmHg) at 24 hrs | | 124.1 ± 4.50 | 102.3 ± 4.74 ** |
| Diastolic BP (mmHg) at 24 hrs | | 59.55 ± 2.00 | 55.63 ± 1.36 |
| Mean Arterial Pressure (mmHg) at 24 hrs | | 77.64 ± 1.82 | 71.63 ± 1.96 * |
| Respiratory rate (No. of breaths/min) | | 20.36 ± 0.88 | 22.00 ± 1.30 |
| LVEF (%) (Median) | | 59.63 ± 1.78 | 57.86 ± 2.14 |
| Fluid balance change (At end of CPB) | | | |
| Positive (%) | | 54.55 | 75.0 |
| Negative (%) | | 36.36 | 25.0 |
| Zero (%) | | 9.09 | 0 |
| Duration of ventilation (hrs) | CMV | 12.30 ± 1.10 | 18.13 ± 3.07 |
| | CPAP | 2.37 ± 0.46 | 22.25 ± 16.59 ** |
| ICU Stay (days) | | 4.50 ± 0.61 | 10.63 ± 4.46 |
| Hospital stay (days) | | 9.81 ± 0.90 | 23.25 ± 5.9* |
| Mortality (number and %) | | 0 & 0% | 2 & 10.5% |
| **Biochemical measurements** | | | |
| Urea (mg/dl) | | 33.27 ± 5.34 | 36.86 ± 3.77 |
| Creatinine (mg/dl) | | 0.73 ± 0.08 | 0.96 ± 0.09* |
| EGFR (ml/min/1.73m2) | | 116.9 ± 7.41 | 93.32 ± 12.74 |
| Hemoglobin (%) | | 10.03 ± 0.40 | 10.33 ± 0.40 |
| ESR (mm/hr) | | 53.60 ± 7.58 | 40.00 ± 9.33 |
| WBC (103/mm3) | | 14.02 ± 2.06 | 20.62 ± 2.82 * |
| Lymphocytes (%) | | 10.55 ± 2.08 | 5.12 ± 0.85 * |
| Monocytes (%) | | 7.09 ± 1.03 | 7.12 ± 1.00 |
| Neutrophils (%) | | 80.18 ± 3.46 | 86.88 ± 1.69 |
| Platelet count (103/mm3) | | 187.2 ± 11.23 | 192.6 ± 39.17 |
| **Post-operative risk score (ICU score)** | | | |
| SOFA Score (Median score)/estimated % Mortality | | 7/18.2 | 8/26.3 |
| APACHE II Score (Median score)/estimated % Mortality | | 9/7.6 | 13/7.75 |

EGFR—Estimated Glomerular Filteration Rate; ESR—Erythrocyte Sedimentation Rate; WBC—White Blood Cells; CMV—Continuous mandatory ventilation; CPAP—Continuous positive airway pressure; ACEI—Angiotensin Converting Enzymes inhibitor; ARB—Angiotensin II receptor blocker;LVEF—Left ventricular ejection fraction; SOFA Score-Sequential organ Failure Assessment II;APACHE II -Acute physiology and Chronic Health Evaluation II.Data expressed in Mean ± S.E.M unless otherwise specified. LVEF, Hemoglobin and Death Rate are expressed in %.

*p<0.05 and

**p<0.01.

electrolytes); oxygenation parameters (PaO2, FiO2 and mechanical ventilation); and hematology (Hematocrit, WBC). The value signifying the most extreme deviation from the normal for each physiological variable within a 24 hour period post operation is used for scoring.

As shown in Table 1, the pre-operative risk score, EuroSCORE II predicts twice the estimated % mortality for group 2 (3.05%) compared to group 1 (1.25%), indicating the presence

of higher risk for group 2 subjects in developing clinical complications upon cardiac surgery. The post-operative ICU score SOFA II indicate higher % mortality (26.3%) in group 2 compared to group 1 (18.2%) while the APACHE II score showed similar score values and estimated % mortality among the groups (Table 2). This establishes the need to carry out studies which determine biomarkers, especially vascular markers which can provide independent predictions, and may also complement risk scores to determine the severity of post-operative clinical outcomes.

## 3.2 Circulating levels of progenitor cells

Changes in the circulating levels of progenitors were enumerated in the individuals using flow cytometry. As can be seen in Fig 3A, for the combined 19 on-pump cases, we observed an early increase in CD34$^+$ cells followed by CD133$^+$ and VEGFR2$^+$ cells. The CD34$^+$ cells count peaked at 1 hour and remained elevated at 6 hours (a 25-fold increase compared to post induction), but gradually decreased by 24 hours (Fig 3A). There was a 15-fold increase in CD133$^+$ cell counts which peaked at 6 hours compared to post induction, later dipping at 24 hours (Fig 3A). In contrast, VEGFR2$^+$ cells had a late response to CPB, which increased gradually post-CPB from 6 hours onwards and remained high at 24 hours (statistically non-significant) (Fig 3A). With regard to double positive cells, EPCs rose from 1 hour onwards and remained elevated at 24 hours (Fig 3B) whereas HSC peaked at 6 hours but dipped by 24 hours (Fig 3B). CD133$^+$VEGFR2$^+$ cells did not show any observable change in numbers upon CPB in the study subjects.

## 3.3 Circulating levels of progenitor cells and levels of lactate in G1 versus G2 subjects

We then sought to determine how changes in circulating progenitors differed between G1 and G2. As can be seen from Fig 4A and 4B, the G1 individuals had an early increase in the number

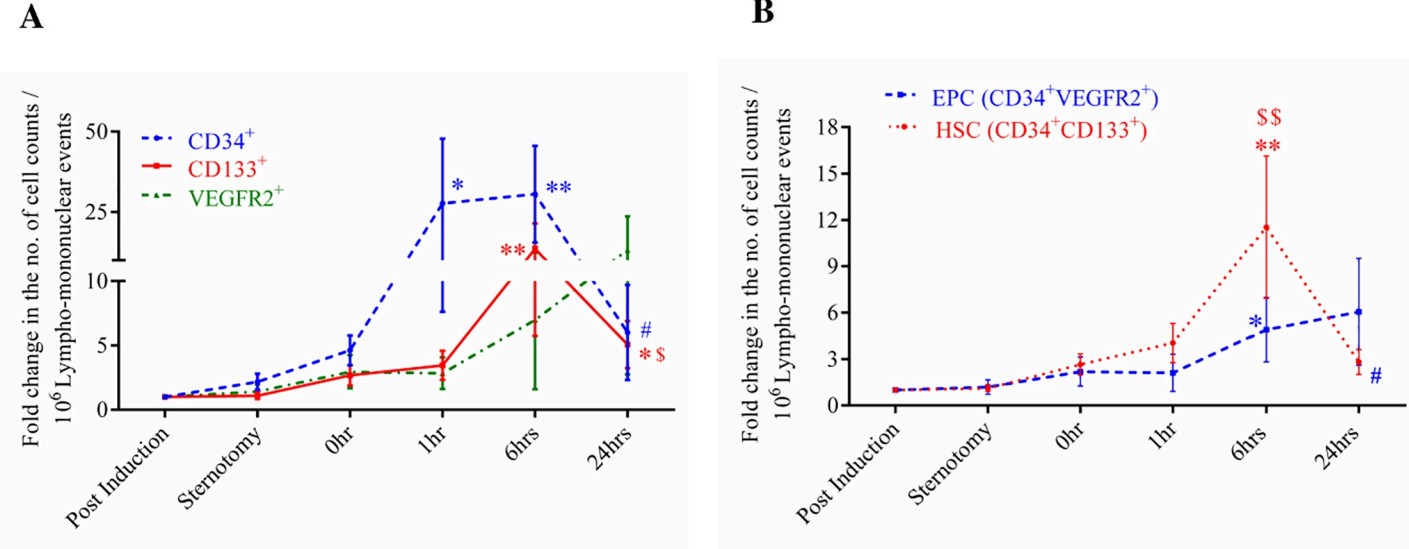

**Fig 3. Fold changes in circulating counts of measured progenitor cells for all 15 on pump cases (4 were considered outlier by statistical package for the social sciences).** (A) Line graph showing fold change in single positive cell counts for CD34$^+$, CD133$^+$& VEGFR2$^+$ cells and (B) Line graph summarizing changes in double positive cell counts for HSCs (CD34$^+$CD133$^+$) and EPCs (CD34$^+$VEGFR2$^+$). Data is plotted as mean ± SEM. $^*$p ≤ 0.05, $^{**}$ p ≤ 0.01 vs post induction, $ p ≤ 0.05, $$ p ≤ 0.01 vs sternotomy and # vs 6 hours. Mann-Whitney's test and Wilcoxon matched pair test were employed accordingly.

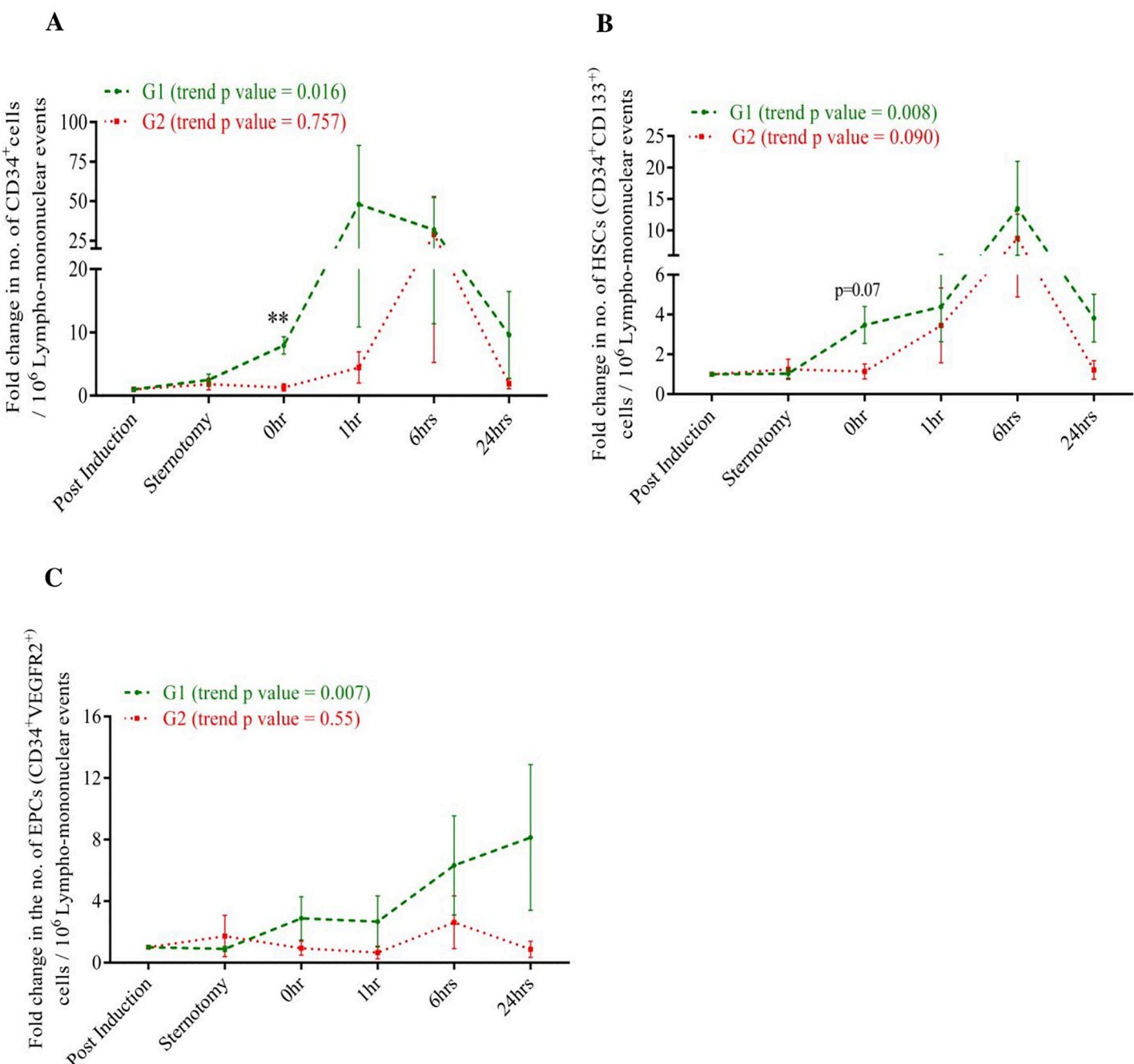

**Fig 4. Circulating counts of progenitor cells in individuals undergoing CPB at various intra-operative time points in G1 and G2 study individuals.** Line graph showing fold change in cell counts for (A) CD34$^+$ (B) HSC (CD34$^+$CD133$^+$) and (C) EPC (CD34$^+$VEGFR2$^+$) in vasoplegic groups namely G1 and G2. Data is plotted as mean ± SEM. $^*p \leq 0.05$, $^{**} p \leq 0.01$ vs G2 at that point. Mann-Whitney's test and Wilcoxon matched pair test were employed accordingly.

of CD34$^+$ from 0 hour onwards, whereas this increase was delayed in G2 individuals. In G2 individuals, an increase in the cell number with regard to CD34$^+$ and HSCs was observed only at 6 hours. EPCs started rising from 1 hour onwards in G1 individuals and remained elevated till 24 hours. This response was blunted in G2 (Fig 4C). The other cell types, i.e. VEGR2$^+$ cells and CD133$^{+,}$ did not show any statistically significant trend for G1 and G2 subjects.

### 3.4 Assessing vasoplegia grouping for individual patients using machine learning

The log-fold change of CD34$^+$ marker for each patient was plotted on the y-axis at different time points with respect to three baselines: induction (B1), sternotomy (B2) and 0 hour (P0) against the case number shown arranged in increasing order of amount of nor-epinephrine dosage given to patients on the x-axis (as in Fig 5A–5C respectively). We observed that, on using CD34$^+$ as a marker, the 0-hour log fold change with respect to baselines B1 i.e induction

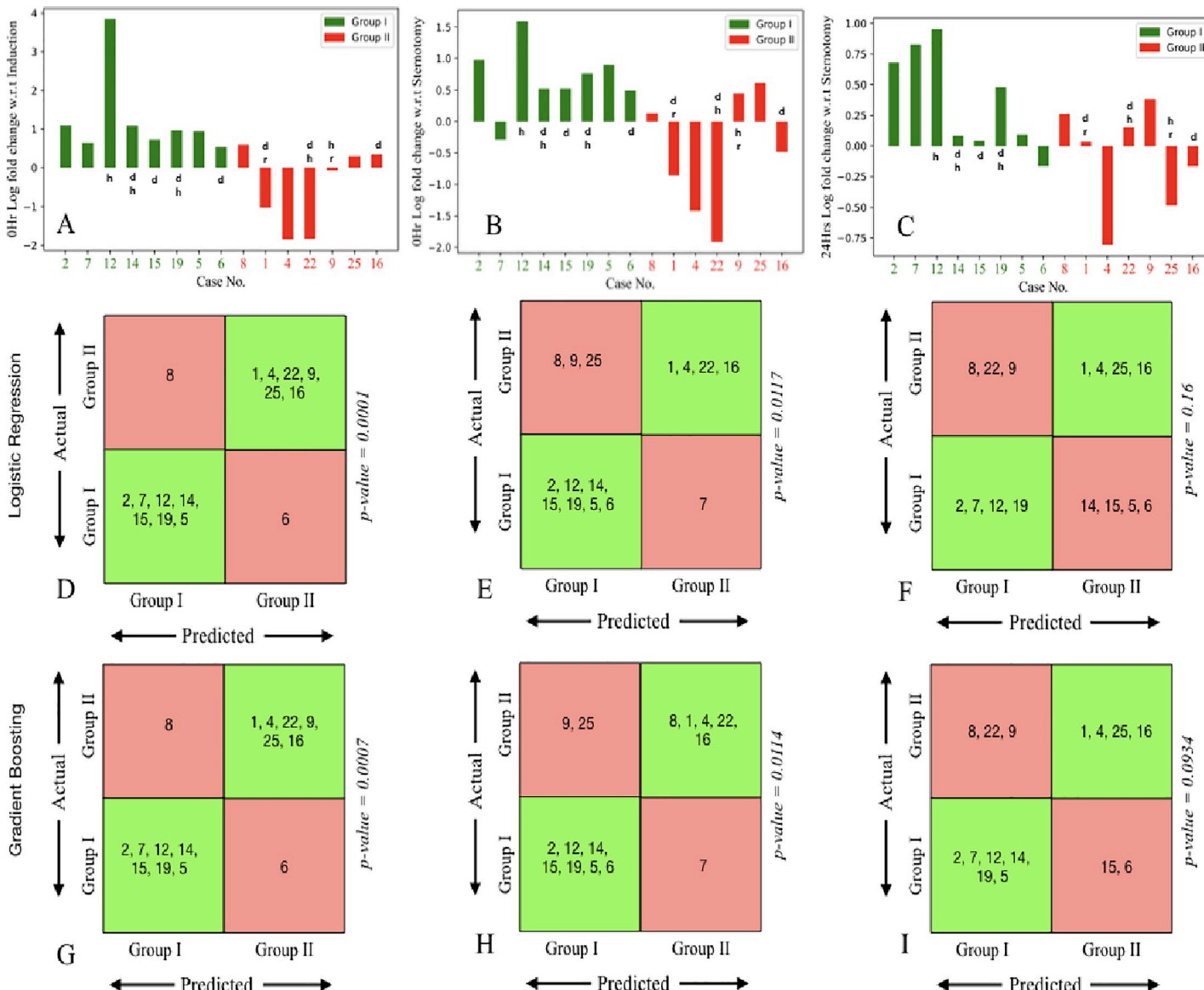

**Fig 5. Bar graph showing log fold change of CD34$^+$ marker for each patient, identified by case number, computed at different time points with respect to three baselines: Induction, sternotomy and 0 hr.** (A) Log fold change at on-pump 0hr with respect to induction. (B) Log fold change for on-pump 0hr time point with respect to sternotomy. (C) Shows 24hr log fold change with respect to sternotomy. Case numbers for patients in G1 (insignificant vasoplegia) are shown in green and those for patients in G2 (significant vasoplegia) are shown in red. The status of individual patients as diabetic (d), hypertensive (h) or having renal failure (r) is also indicated together with each bar. (D), (E), (F) show machine learning groupings for G1 and G2 using logistic regression and (G), (H), (I) show the same based on gradient boosting. Correctly classified patients are shown, by case number, in green boxes, and misclassified patients in red boxes. Corresponding p-values are shown.

**Table 3. Power analysis.**

|  | 0 hour vs induction | 0 hour vs sternotomy | 24 hour vs sternotomy |
|---|---|---|---|
| Mean I (Non-Vasoplegic) | 1.23 | 0.68 | 0.37 |
| Mean II (Vasoplegic) | -0.51 | -0.50 | -0.09 |
| Standard deviation I | 1.08 | 0.53 | 0.42 |
| Standard deviation II | 1.05 | 0.96 | 0.43 |
| Pooled Std Deviation | 1.06 | 0.76 | 0.42 |
| Cohen's $d$ effect size | 1.63 | 1.56 | 1.10 |
| Power (actual effect size) | 0.62 | 0.55 | 0.20 |
| Power (effect size 1) | 0.20 | 0.20 | 0.20 |
| Power (effect size 1.5) | 0.55 | 0.55 | 0.55 |
| Power (effect size 2) | 0.77 | 0.77 | 0.77 |

(Fig 5A) and B2 i.e sternotomy (Fig 5B) and 24 hours fold change with respect to baseline B2 (Fig 5C) show the clearest visual trend in representing the difference between G1 and G2.

Patients were then classified using logistic regression (Fig 5D–5F) and gradient boosting (Fig 5G–5I) algorithms on the above-mentioned log fold change values for the CD34$^+$ marker. Grouping on each patient were made by leaving that patient out and training on other patients. Gradient boosting outperformed logistic regression in general, with both methods mis-classifying only two patients in the best case (Fig 5D and 5G). The performance of the classifiers was significant ($p<0.05$) for Fig 5D, 5E, 5G and 5H (0 hour compared to baselines B1 and B2).

**Power analysis.** For each time point, we calculated the sample means and standard deviations separately for the two groups (I = non-vasoplegic or insignificantly vasoplegic; II = vasoplegic), and the pooled standard deviation. We calculated the effect size given as Cohen's d = (difference in means)/(pooled standard deviation). We calculated the statistical power for the observed effect size, as well as for effect sizes of 1.0, 1.5 and 2.0, at a desired significance of 0.05 for 8 non-vasoplegic and 7 vasoplegic patients, by simulating 100 data sets per effect size. For the 0-hour vs induction and 0 hour vs sternotomy time points, we find a large effect size and a moderately high statistical power, as shown in Table 3 below.

**Propensity matching.** Given the small dataset, a thorough propensity matching on all parameters is unfeasible. Instead, a Euroscore II value was calculated for each patient based on their clinical parameters, and these were matched. Out of eight non-vasoplegic and seven vasoplegic patients, we found a subset of five good matches per group, as given in the supplementary table (S1 Table) and supplementary figure (S2 Fig). Using these matched subsets of 5 vs 5 patients led to comparably good predictions as with the full set. Since the Euroscore II is comparable between the two groups, there is no significance difference in the risk profiles between the two groups.

## 4. Discussion

In this study, we report differences in time dependent changes in levels of circulating progenitors and EPCs in CPB surgery in individuals exhibiting clinically significant and insignificant vasoplegia. We have demonstrated (as in Fig 3A and 3B) that CD34$^+$ cells are the first one to respond to CPB, followed by CD133$^+$, HSCs, EPCs and VEGFR2$^+$ cells. This part of the study supports the conclusions of previous studies investigating the mobilization of progenitor cells in circulation during and after CPB. Our findings on CD34$^+$ and HSCs are comparable with previous studies done by Dotsenko *et al* and Scheubel *et al* [28,29]. They reported an increase in CD34$^+$ cell counts at the end of surgery which contained to remain elevated 24 hours postoperation. Similarly, the rise in circulating HSCs peaks at 6 hours post-operatively and

decreases after 24 hours. Increased EPCs and VEGFR2+ levels at the end of the surgery have also been reported in previous studies [30–32]. Notably, the kinetics of progenitor cells and lactate levels follow a trend similar to that seen in Figs 2B, 3A and 3B, which suggests hypoxia driven mobilization of progenitor cells [33].

To the best of our knowledge this study represents the first attempt to associate changes in numbers of progenitors with the severity of vasoplegia. We have shown that individuals who do not develop clinically significant vasoplegia after surgery (G1) showed significant early increase in the number of CD34+ and HSCs. This rise was not observed in the individuals who developed significant vasoplegia (G2). Similarly, a blunted response of EPCs was observed in G2 individuals in comparison to G1.

Upon categorization based on the severity of vasoplegia, the early response of CD34+ and HSC counts in G1, as seen in Fig 4A and 4B, might indicate the body's attempt to limit vascular damage by inducing repair mechanism against the insult caused due to CPB. This early response is compromised in G2 individuals. Prior studies have claimed that CD34+ cells promote therapeutic angiogenesis through paracrine signaling in response to myocardial ischemia and have the potential to ensure engraftment of transplanted cells [34]. Recently, Cogle *et al* have shown that the increase in the number of CD34+ cells in bone marrow is directly associated with improved functional outcomes after acute myocardial infarction (AMI) in the Timing in Myocardial Infarction Evaluation (TIME) and Late TIME clinical trials [35]. In another study, Fadini *et al* have found that CD34+ cells have a strong negative correlation with cardiovascular risk [36]. Similarly, HSCs are known to promote neovascularization on tissue engineered construct and have shown significant correlation with autologous white cell count and engraftment kinetics in myeloma patients [37,38]. Based on these studies and our observations, we speculate that endothelial injury caused by CPB, if not accompanied by an early mobilization of sufficient circulating progenitor cells, may result in impaired vascular repair, thereby contributing to the severity of vasoplegia. However, the reason for delayed response of CD34+ and HSCs cells in G2 individuals is still unknown. Given the variability of vasoplegia even among patients within different risk groups, it would be reasonable to assume that pre-operative profiles may influence the ability to mobilize circulating progenitors and HSCs.

We also observed a depression in EPC count in G2 patients as seen in Fig 4C. The reason for decreased circulating counts of EPCs in G2 could be due to compromised migration, homing and reparative potential of these cells as shown by Ruel *et al*. They demonstrated that the migratory effect of EPC was compromised in on-pump CPB patients [32]. Further, Lei Du *et al* have reported that EPCs start homing to the injured site after 4 hours from the end of the surgery, reporting a drop in the counts of circulating EPCs after 4 hours accompanied by a corresponding increase in their numbers in lung and kidney tissues [39].

To support our findings, we employed machine learning techniques to evaluate our categorization of groups G1 and G2 with respect to the CD34+ marker. We found that the log fold change values of CD34+ markers (as seen in Fig 5A–5C) reflected our findings in terms of an increasing trend in G1 and decreasing trend in G2. Among the algorithms we utilized, namely logistic regression and gradient boosting, we found that gradient boosting performed better in classifying patients into G1 and G2 as seen in Fig 5G–5I. We found that in most patients, the clinical classification agreed with the grouping from gradient boosting. However certain patients, namely patients 6, 8 and 9 in our study, were frequently misclassified. Of these, patient 9 had Pre-CPB renal failure, while patients 6 and 8 had borderline levels of vasoplegia as indicated by norepinephrine dosage. While these results are promising, a study on a larger cohort would enable us to refine our predictive methods for better understanding.

In summary, this pilot study shows that a distinctive pattern of EPC response characterizes patients likely to develop clinically significant vasoplegia. Further studies will help clarify the

profile and the use of the EPC response as a biomarker to dictate the threshold to start pressors (or low dose steroids).

## Study limitations

There are certain limitations of the current study. For instance, due to small sample size, certain observations did not reach statistical significance. The other major limitation of the study is the non-availability of the blood sample beyond 24 hours. This limited us to observations of the kinetics of progenitor cells only up to 24 hours. Additionally, we did not measure the levels of secretory endothelial markers during perioperative time points to assess endothelial damage. Further studies with large cohorts tracking the changes both between and within risk stratified groups undergoing cardiopulmonary bypass will be needed to determine if a particular response profile could be used to predict the occurrence of significant vasoplegia.

## Supporting information

**S1 Fig. Gating strategy applied for whole blood.**
(TIF)

**S2 Fig. Propensity matched subgroups of 5 patients under each group (Group I and Group II).**
(TIF)

**S1 Table. Propensity matching.**
(TIF)

## Acknowledgments

The authors wish to acknowledge The Medical Team (Surgeons, Operation Theatre and ICU staff), Department of Cardiothoracic Surgery, Apollo Hospitals, Chennai for coordination in sample collection. We acknowledge Mr. Emmanuel, Senior Perfusionist for his technical assistance.

## Author Contributions

**Conceptualization:** Madhulika Dixit, Paul Ramesh Thangaraj.

**Data curation:** Sanhita Nandi, Uma Rani Potunuru, Paul Ramesh Thangaraj.

**Formal analysis:** Sanhita Nandi, Uma Rani Potunuru, Chandrani Kumari, Abel Arul Nathan, Gautam I. Menon, Rahul Siddharthan.

**Funding acquisition:** Madhulika Dixit, Paul Ramesh Thangaraj.

**Investigation:** Sanhita Nandi, Rahul Siddharthan, Madhulika Dixit, Paul Ramesh Thangaraj.

**Methodology:** Sanhita Nandi, Uma Rani Potunuru, Jayashree Gopal, Madhulika Dixit.

**Project administration:** Madhulika Dixit, Paul Ramesh Thangaraj.

**Resources:** Paul Ramesh Thangaraj.

**Software:** Chandrani Kumari, Rahul Siddharthan.

**Supervision:** Jayashree Gopal, Gautam I. Menon, Rahul Siddharthan, Madhulika Dixit, Paul Ramesh Thangaraj.

**Validation:** Jayashree Gopal, Rahul Siddharthan, Paul Ramesh Thangaraj.

**Visualization:** Jayashree Gopal, Paul Ramesh Thangaraj.

**Writing – original draft:** Sanhita Nandi, Abel Arul Nathan, Jayashree Gopal, Rahul Siddharthan, Madhulika Dixit, Paul Ramesh Thangaraj.

**Writing – review & editing:** Sanhita Nandi, Uma Rani Potunuru, Chandrani Kumari, Abel Arul Nathan, Jayashree Gopal, Gautam I. Menon, Rahul Siddharthan, Paul Ramesh Thangaraj.

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
