## [Decision Letter · Decision Letter 0]

27 Jul 2020

PONE-D-20-15232

Altered kinetics of circulating progenitor cells in cardiopulmonary bypass (CPB) associated vasoplegic patients: An observational study

PLOS ONE

Dear Dr. Thangaraj,

Thank you for submitting your manuscript to PLOS ONE. After careful consideration, we feel that it has merit but does not fully meet PLOS ONE’s publication criteria as it currently stands. Therefore, we invite you to submit a revised version of the manuscript that addresses the points raised during the review process.

Specifically, the patient population should be expanded and a control group should be added. Furthermore, more detailed statistical analysis, such as a multivariate analysis and a propensity score matching, are needed to confirm the strenght of the results.

We look forward to receiving your revised manuscript.

Kind regards,

Antonio Cannatà

Academic Editor

PLOS ONE

Journal Requirements:

Reviewers' comments:

Reviewer's Responses to Questions

**Comments to the Author**

1. Is the manuscript technically sound, and do the data support the conclusions?

Reviewer #1: Partly

Reviewer #2: Partly

Reviewer #3: Yes

2. Has the statistical analysis been performed appropriately and rigorously? 

Reviewer #1: Yes

Reviewer #2: I Don't Know

Reviewer #3: Yes

3. Have the authors made all data underlying the findings in their manuscript fully available?

Reviewer #1: Yes

Reviewer #2: Yes

Reviewer #3: Yes

4. Is the manuscript presented in an intelligible fashion and written in standard English?

Reviewer #1: Yes

Reviewer #2: Yes

Reviewer #3: Yes

5. Review Comments to the Author

Reviewer #1: Dr. Thangaraj,

We would like to thank you and your colleagues for submitting your work to PLOS ONE. This type of translational research is much welcomed to this journal. Cardiothoracic surgeons constantly deal with the vasoplegic patient after cardiopulmonary bypass and therefore, attempts to finding biomarkers to help determine which patients may have a vasoplegic response can result in many rescued patient. This type of study is particularly difficult to pull off, due to multiple confounding factors regarding patients so I commend you on the attempt. However, there are multiple concerns I have regarding your study.

1. This study is severely limited by the sample size of patients. In order to make any meaningful generalizations, sample sizes of 11 and 8 are inadequate. Please do a power analysis if possible.

2. The patient population is not adequately described in multiple areas. There is a range for the ages of the patients which is 40-70. This is a very wide range and there are multiple studies showing that age affects inflammation. The study has medication intake per group, but statins are not included and statins are showed to be anti-inflammatory. Lopressor is a very popular medication which has also been shown to be anti-inflammatory. Also, preoperative comorbidities like COPD, chronic liver disease, steroid use, etc can have such a major impact on outcome. Did any of the patients have heart failure? What was the PREOPERATIVE LVEF of the patients? Was there pulmonary artery hypertension or RV dysfunction?

At this point, I have no clue what types of patients are being operated on.

3. Please use a score to determine risk assessment of the cardiac surgery patients. This will give us an idea of how sick the patients are that were included in the study. In the US, we use a the STS-PROM. Whatever score is suitable for you, please include. If could very well be the patients who had an increase pro-inflammatory response were patients who were sicker prior to surgery.

4. Operative details were left out. What type of operations did these patients undergo? I don't believe a 40 year old undergoing a CABG and a 70 year old undergoing CABG will react differently. A 70 year old is expected to have level of CAD, while a 40 year old is likely to not have CAD. I would assume the 40 year old may have increased proinflammatory markers prior to surgery, does this affect your study? What about a 40 yo undergoing a MVrepair for flailed mitral disease whose relatively healthy vs. a 40 year old with endocarditis? We see these patients all the time.

5. More on operative detail - What were the CPB times for the patients? What was the cross-clamp times? What cardioplegia did you use? What MAPs were maintained in the OR? What was the flow for the patients? All of these things that happen in the OR affects pro-inflammatory markers and lactic acidosis, etc. If you have a patient undergoing a cross-clamp time of 180 minutes vs. 80 minutes, I would assume that the longer cross clamp time would have more pro-inflammatory markers. We certainly see that these patients are usually sicker coming out of the operating room. You may want to look into the affects of CPB time or cross clamp time on the markers you discussed.

6. Post-operative course is not described passed 24 hours. First, the study has a 10% mortality rate, which is exceedingly high depending on the operation and how sick the patient is prior to surgery. Once again, this is a reason patient population needs to be more adequately described and a risk assessment score needs to be included in the paper.

7. Norepi is the drug that was used as a consensus for clinical vasoplegia. How was it determined to place this patient on norepi vs. epi. What about norepi vs. vasopressin? What about clinical fluid status? Maybe the patient is not vasoplegic, but dry and required fluids? Once again, patient population needs to be described.

Once again, I commend you on a very difficult study. The results of the study were definitely interesting. However, without the information stated above, I cannot agree that there is applicability to these findings. There are many pro- and anti-inflammatory aspects of the patients that are confounding your study. A multivariable analysis may need to be done. I am happy to re-review the study once more information and work is done. I encourage the authors to look into some of my suggestions as I believe it would make this work clinically more applicable.

Reviewer #2: The subject of the paper is scientifically relevant and the the work performed correctly, as to my expertise. Nevertheless, the population enrolled to the study is small, considered the following sub-categorization in two groups. The differences observed by the authors wil certainly be reinforced by inclusion of 10-20 more patients and scientific soundness of the paper would benefit from including other groups of patients - off-pump cardiac surgery as a "control group" and ECMO patients.

Reviewer #3: I congratulate the author for the great job.

The authors analysed vasoplegia observed post-cardiopulmonary bypass and the correlation with CD34+ cells and CD34+CD133+ hematopoietic stem cells.

This is an important topic in cardiac surgery, source of important comorbidity and post-operative mortality.

The paper is well written and the conclusion are definitely supported.

Regards

6. PLOS authors have the option to publish the peer review history of their article (what does this mean?). If published, this will include your full peer review and any attached files.

Reviewer #1: No

Reviewer #2: No

Reviewer #3: **Yes: **Cristina Barbero, MD, PhD

---

## [Author Response · Author response to Decision Letter 0]

23 Sep 2020

Response to Reviewers

Editor comment: Furthermore, more detailed statistical analysis, such as a multivariate analysis and a propensity score matching, are needed to confirm the strength of the results. 

Ans: Given the small data set size, it is unfeasible to perform propensity matching on a large number of parameters. We have instead performed a basic propensity matching of the two groups of patients using Euroscore II (or Euro SCORE II) values, and find similar results to the full set. Results are shown in a separate heading under propensity matching under section 3.4 and included in the revised manuscript as Supplementary table 1, (S1 Table) and supplementary figure S2 (S2_fig).

Reviewer #1

Query #1: This study is severely limited by the sample size of patients. In order to make any meaningful generalizations, sample sizes of 11 and 8 are inadequate. Please do a power analysis if possible.

Ans: The objective of this study was to observe if the endothelial repair response as indicated by change in circulating progenitor cell mobilization was different in patients with and without clinically defined vasoplegia. In that regard the fold change experienced was by individual patients with respect to their own baseline. The subsequent trajectory of the response was characterized by both quantitative (change in number of circulating progenitor cells) and qualitative (change in types of circulating progenitor cell markers) methods. The analysis was made to see if this quantum varied between patients who developed vasoplegia and those that did not (Hence, each patient serves as their own control). If there were indications that it was the case then we planned to do a bigger study as this was more in the nature of a pilot study. 

Power analysis: 

We have performed a power analysis which is now included in results section 3.4. The effect size (Cohen's d) for our data is 1.63 (i.e., the means of the two groups are 1.63 standard deviations apart), which is a large effect. Based on simulating random data with the same distribution (same effect size), for 8 non-vasoplegic and 7 vasoplegic patients at a significance threshold of 0.05, the statistical power is 0.6. The results of the power analysis are shown in table-3 in the revised manuscript. 

Query #2: The patient population is not adequately described in multiple areas. There is a range for the ages of the patients which is 40-70. This is a very wide range and there are multiple studies showing that age affects inflammation. The study has medication intake per group, but statins are not included and statins are showed to be anti-inflammatory. Lopressor is a very popular medication which has also been shown to be anti-inflammatory. Also, preoperative comorbidities like COPD, chronic liver disease, steroid use, etc. can have such a major impact on outcome. Did any of the patients have heart failure? What was the PREOPERATIVE LVEF of the patients? Was there pulmonary artery hypertension or RV dysfunction? At this point, I have no clue what types of patients are being operated on.

Ans: We partly agree with the reviewer. As we have given most critical information required to understand the patient population which might affect the post CPB outcomes (Please refer to Table-1 in the revised manuscript). We felt that ACEI (Angiotensin-converting-enzyme inhibitor) and ARBs (Angiotensin II receptor blockers) have most significant effect on post-operative vasodilatory syndromes; however, as suggested by the reviewer, we have now added the details of intake of beta blocker (e.g. Lopressor a brand name of Metoprolol in USA, which is selective β1 receptor blocker) as well as steroids and statins in table 1 of the revised manuscript.

We have included the pre OP co-morbidities heart failure, RV dysfunction and PAH as suggested by the reviewer. However, the idea was to see if prediction of vasoplegia from EPC behavior alone was possible and if so, how many patients could be accurately identified as developing vasoplegia for a given EPC profile independent of co-morbidity or risk score profile.

Moreover, in the revised manuscript we have included objective risk scores which takes in to account the pre-operative co-morbidities (Please refer to answer to query # 3 for more details).

Query #3: Please use a score to determine risk assessment of the cardiac surgery patients. This will give us an idea of how sick the patients are that were included in the study. In the US, we use the STS-PROM. Whatever score is suitable for you, please include. If could very well be the patients who had an increase pro-inflammatory response were patients who were sicker prior to surgery.

Ans: We have now included risk assessment scores Euro SCORE II, SOFA II and APACHE II used in our hospital set up in table 2 of result section 3.1 in the revised manuscript. The results of the score calculation is appropriately discussed in the revised manuscript. Euro SCORE II (European System for Cardiac Operative Risk Evaluation II) is used as a pre-operative indicator of patients’ risk profile[1,2]. SOFA II (Sequential organ Failure Assessment II) is used to determine level of organ dysfunction and mortality risk in ICU patients. APACHE II (Acute physiology and Chronic Health Evaluation II) score estimates ICU mortality[3–5]. From Euro SCORE II calculation it was observed that the group 2 subjects had two-fold % estimated mortality prediction compared to group 1. Post-operative ICU score, SOFA II indicates higher % mortality (26.3%) in group 2 compared to group 1 (18.2%) and APACHE II showed similar estimated % mortality for group 1 (7.6%) and group 2 (7.75%) (Table-2). Hence, the enumeration and identification of circulating progenitors in our study subjects in a temporal manner could add new parameter in the prediction of the severity of post CPB clinical state.

Query #4. Operative details were left out. What type of operations did these patients undergo? I don't believe a 40-year-old undergoing a CABG and a 70-year old undergoing CABG will react differently. A 70-year old is expected to have level of CAD, while a 40-year old is likely to not have CAD. I would assume the 40-year-old may have increased pro-inflammatory markers prior to surgery; does this affect your study? What about a 40-year undergoing a MV repair for failed mitral disease whose relatively healthy vs. a 40-year-old with endocarditis? We see these patients all the time. 

Ans: We have now included the type of operation conducted in the subject group in table 1 of the revised manuscript as asked by the reviewer. 

Patient characteristics like pre-operative co-morbidities and operation related factors were discussed while calculating the objective risk scores Euro SCORE II. We have now included pre-operative risk prediction score Euro SCORE II in the revised manuscript (Table-1).

Query #5: More on operative detail - What were the CPB times for the patients? What were the cross-clamp times? What cardioplegia did you use? What MAPs were maintained in the OR? What was the flow for the patients? All of these things that happen in the OR affects pro-inflammatory markers and lactic acidosis, etc. If you have a patient undergoing a cross- clamp time of 180 minutes vs. 80 minutes, I would assume that the longer cross clamp time would have more pro- inflammatory markers. We certainly see that these patients are usually sicker coming out of the operating room. You may want to look into the effects of CPB time or cross clamp time on the markers you discussed. 

Ans: We do agree with the reviewer that longer cross clamp time would have more pro-inflammatory markers. Other intra-operative parameters such as CPB time, flow rates and MAPs too could affect the patient outcome. We have now included duration of CPB and Cross clamp time as well as flow rate in table 2. However, we did not find significant difference in these parameters among the 2 groups as shown in Table-2 of the revised manuscript. 

Blood cardioplegia was used for all the cardiac cases. The cardioplegic solution was a mixture of patient’s own blood and crystalloid solution. The crystalloid solution was a mixture of Plasmalyte (500 mL) and St. Thomas' solution (20 mL). The delivery was done by the use of a specialized roller pump (Sarns Medical Systems) at the ratio of 4:1 where 4 parts of blood and 1 part of the mixed cardioplegia solution was used. In lung transplant the lung preservative used was Perfadex solution (Medisan, Uppsala, Sweden) containing low-potassium (K+ 6 mmol/L) and extracellular electrolytes (Na+ 138 mmol/L; Cl- 142 mmol/L; Mg++ 0.8 mmol/L; Dextran 40 g);H2PO4- 0.8 mmol/L; Glucose (0.91 g); Osmolarity (292 mOsm/L). All these details are now included in methods section 2.2 of the revised manuscript.

The MAP of 60 to 80 mmHg was maintained in the OR for all the surgery cases which were included in our study group. 

We examined this question and found no correlation between CPB time or cross clamp time with the markers. This is evident in the bar plot shown here for the reviewers' benefit. (Shown in the attached file ' Response to Reviewers)

Query #6: Post-operative course is not described passed 24 hours. First, the study has a 10% mortality rate, which is exceedingly high depending on the operation and how sick the patient is prior to surgery. Once again, this is a reason patient population needs to be more adequately described and a risk assessment score needs to be included in the paper. 

Ans: Vasodilatory syndrome post cardiac surgery occurs between 6 to 24 hour. The study was designed to observe circulating progenitor cell response during the operation and in the first 24 hours. However, we have included post-operative milestones such as total hours ventilated and duration of hospital stay. 

The total number of cardiac surgeries carried out during the period of study was 260, out of which 2 patients died. Hence; the overall mortality rate during the period of study is 0.77%. Out of total 260 cardiac surgeries 19 patients consented to be part of the pilot study of which 2 patients died. Thus, the 10% mortality was a reflective of number of subjects consented and enrolled in the study. As mentioned in the reply to query #3, we have included the risk score in the result section of the revised manuscript for the study population.

Query #7: Norepi is the drug that was used as a consensus for clinical vasoplegia. How was it determined to place this patient on norepi vs. epi. What about norepi vs. vasopressin? What about clinical fluid status? Maybe the patient is not vasoplegic, but dry and required fluids? Once again, patient population needs to be described. 

Ans: The protocol in our department is use of noradrenalin limited to patients who have moderate ejection fraction (30-50%) or normal EF (> 50%) if there is hypotension (< 60mmHg MAP), after the CVP was restored to pre surgical (at induction level) or as identified by the surgeon or anesthetist as having been the optimal filling in the OR.

In poor EF (<30%) adrenaline or dobutamine is first line and nor-adrenaline added only if warm peripheries and low systemic vascular resistance index (SVRI<1600) is documented. Increase in noradrenalin above 5mls/kg/min is not allowed unless further low SVRI by PAFC (Pulmonary artery floatation catheter) reading and good LV and RV function by echo is documented. Vasopressin is added in post-operative period only if the noradrenalin dose exceeds 10mls/kg/min. 

Clinical fluid status difference between two groups is added in the revised manuscript. The results are also shown as percentage of subjects with positive, negative or zero fluid state at the end of CPB in Table-2 of the revised manuscript. 

Reviewer #2:

Query #1: The subject of the paper is scientifically relevant and the work performed correctly, as to my expertise. Nevertheless, the population enrolled to the study is small, considered the following sub-categorization in two groups. The differences observed by the authors will certainly be reinforced by inclusion of 10-20 more patients and scientific soundness of the paper would benefit from including other groups of patients off-pump cardiac surgery as a "control group" and ECMO patients.

Ans: The study was designed to elucidate if there was a particular trajectory to the EPC response that was associated with the development of vasoplegia irrespective of patient profile. Our intention was not to compare different patient sub groups but to compare the circulating progenitor cell response profiles among the groups who developed vasoplegia to those who did not.

We would like to bring in the kind attention of the reviewer that the patient’s own baseline/post induction blood sample acts as patient’s control. This helps us to develop a patient’s EPC profile which is comparable between the patients. Therefore, a separate control group for comparisons is not necessary for our study design. 

Reviewer #3: 

I congratulate the author for the great job. The authors analysed vasoplegia observed post-cardiopulmonary bypass and the correlation with CD34+ cells and CD34+CD133+ hematopoietic stem cells. This is an important topic in cardiac surgery, source of important co morbidity and post-operative mortality. The paper is well written and the conclusions are definitely supported.

Ans: We thank the reviewer for the comments. 

REFERENCES

1. Roques F, Nashef SA, Michel P, Gauducheau E, de Vincentiis C, Baudet E, et al. Risk factors and outcome in European cardiac surgery: analysis of the EuroSCORE multinational database of 19030 patients. Eur J cardio-thoracic Surg Off J Eur Assoc Cardio-thoracic Surg. 1999;15: 813–816. doi:10.1016/s1010-7940(99)00106-2

2. Roques F, Michel P, Goldstone AR, Nashef SAM. The logistic EuroSCORE. European heart journal. England; 2003. pp. 881–882. doi:10.1016/s0195-668x(02)00799-6

3. Knaus WA, Draper EA, Wagner DP, Zimmerman JE. APACHE II: a severity of disease classification system. Crit Care Med. 1985;13: 818–829. 

4. Wong DT, Knaus WA. Predicting outcome in critical care: the current status of the APACHE prognostic scoring system. Can J Anaesth. 1991;38: 374–383. doi:10.1007/BF03007629

5. Mei YQ, Ji Q, Liu H, Wang X, Feng J, Long C, et al. Study on the relationship of APACHE III and levels of cytokines in patients with systemic inflammatory response syndrome after coronary artery bypass grafting. Biol Pharm Bull. 2007;30: 410–414. doi:10.1248/bpb.30.410

---

## [Decision Letter · Decision Letter 1]

14 Oct 2020

PONE-D-20-15232R1

Altered kinetics of circulating progenitor cells in cardiopulmonary bypass (CPB) associated vasoplegic patients: An observational study

PLOS ONE

Dear Dr. Thangaraj,

Thank you for submitting your manuscript to PLOS ONE. After careful consideration, we feel that it has merit but does not fully meet PLOS ONE’s publication criteria as it currently stands. Therefore, we invite you to submit a revised version of the manuscript that addresses the points raised during the review process.

Specifically, a thorough grammar revision is needed. 

We look forward to receiving your revised manuscript.

Kind regards,

Antonio Cannatà

Academic Editor

PLOS ONE

Additional Editor Comments (if provided):

A thorough grammar revision is needed. Please consider using an external proofreading before resubmitting the manuscript.

Reviewers' comments:

Reviewer's Responses to Questions

**Comments to the Author**

1. If the authors have adequately addressed your comments raised in a previous round of review and you feel that this manuscript is now acceptable for publication, you may indicate that here to bypass the “Comments to the Author” section, enter your conflict of interest statement in the “Confidential to Editor” section, and submit your "Accept" recommendation.

Reviewer #1: All comments have been addressed

Reviewer #2: All comments have been addressed

2. Is the manuscript technically sound, and do the data support the conclusions?

Reviewer #1: Yes

Reviewer #2: Yes

3. Has the statistical analysis been performed appropriately and rigorously? 

Reviewer #1: Yes

Reviewer #2: Yes

4. Have the authors made all data underlying the findings in their manuscript fully available?

Reviewer #1: Yes

Reviewer #2: Yes

5. Is the manuscript presented in an intelligible fashion and written in standard English?

Reviewer #1: Yes

Reviewer #2: Yes

6. Review Comments to the Author

Reviewer #1: Thank you for revising your manuscript.

I do very much appreciate the fact that a lot of the data that I requested was added to the manuscript. I think when you talk about vasoplegia, it is important to also have a good clinical picture for such a picture. With regards to this, I think I would like to see more about what to do with this information in a more clinical setting? Do we prophylactically treat these patients who are predisposed to vasoplegia? Please add something short regarding what to do with your information.

Also, please review grammar and english syntax.

I commend the authors on an interesting study and a well written paper.

Reviewer #2: The authors correctly addressed my comments and those of the other reviewer. The reason of enrolling such a small population of patients is now better outlined and is reasonable. In my opinion, the pilot nature of the study must be clear from the very title of the paper, so my suggestion is to add "a pilot study" to the title.

7. PLOS authors have the option to publish the peer review history of their article (what does this mean?). If published, this will include your full peer review and any attached files.

Reviewer #1: No

Reviewer #2: No

---

## [Author Response · Author response to Decision Letter 1]

28 Oct 2020

Editor comment: A thorough grammar revision is needed. 

Ans: As per the suggestion by the editor, we have made a thorough grammar revision for the revised manuscript.

Reviewer #1

Query #1: Thank you for revising your manuscript.

I do very much appreciate the fact that a lot of the data that I requested was added to the manuscript. I think when you talk about vasoplegia, it is important to also have a good clinical picture for such a picture. With regards to this, I think I would like to see more about what to do with this information in a more clinical setting? Do we prophylactically treat these patients who are predisposed to vasoplegia? Please add something short regarding what to do with your information.

Also, please review grammar and English syntax. 

I commend the authors on an interesting study and a well written paper.

Ans: We thank the reviewer for the comments. We have now included a short summary regarding the findings and perspective of the current pilot study in the discussion section of the revised manuscript. 

Query #2: The authors correctly addressed my comments and those of the other reviewer. The reason of enrolling such a small population of patients is now better outlined and is reasonable. In my opinion, the pilot nature of the study must be clear from the very title of the paper, so my suggestion is to add "a pilot study" to the title.

Ans: We thank the reviewer for the comment. As suggested by the reviewer we have changed the title from ‘Altered kinetics of circulating progenitor cells in cardiopulmonary bypass (CPB) associated vasoplegic patients: An observational study’ to ‘Altered kinetics of circulating progenitor cells in cardiopulmonary bypass (CPB) associated vasoplegic patients: A pilot study’.

---

## [Editor Report · Decision Letter 2]

2 Nov 2020

Altered kinetics of circulating progenitor cells in cardiopulmonary bypass (CPB) associated vasoplegic patients: A pilot study

PONE-D-20-15232R2

Dear Dr. Thangaraj,

We’re pleased to inform you that your manuscript has been judged scientifically suitable for publication and will be formally accepted for publication once it meets all outstanding technical requirements.

Kind regards,

Antonio Cannatà

Academic Editor

PLOS ONE

---

## [Editor Report · Acceptance letter]

9 Nov 2020

PONE-D-20-15232R2 

Altered kinetics of circulating progenitor cells in cardiopulmonary bypass (CPB) associated vasoplegic patients: A pilot study 

Dear Dr. Thangaraj:

I'm pleased to inform you that your manuscript has been deemed suitable for publication in PLOS ONE. Congratulations! Your manuscript is now with our production department. 

Kind regards, 

on behalf of

Dr. Antonio Cannatà 

Academic Editor

PLOS ONE